# Preoperative Mutational Analysis of Circulating Tumor Cells (CTCs) and Plasma-cfDNA Provides Complementary Information for Early Prediction of Relapse: A Pilot Study in Early-Stage Non-Small Cell Lung Cancer

**DOI:** 10.3390/cancers15061877

**Published:** 2023-03-21

**Authors:** A. N. Markou, D. Londra, D. Stergiopoulou, I. Vamvakaris, K. Potaris, I. S. Pateras, A. Kotsakis, V. Georgoulias, E. Lianidou

**Affiliations:** 1Lab of Analytical Chemistry, Department of Chemistry, National and Kapodistrian University of Athens, 15771 Athens, Greece; 2Department of Pathology; ‘Sotiria’ General Hospital for Chest Diseases, 11527 Athens, Greece; 3Department of Thoracic Surgery, ‘Sotiria’ General Hospital for Chest Diseases, 11527 Athens, Greece; 42nd Department of Pathology, Medical School, National and Kapodistrian University of Athens, “ATTIKON” General Hospital of Athens, 12452 Athens, Greece; 5Department of Medical Oncology, University General Hospital of Larissa, 41334 Thessaly, Greece; 6First Department of Medical Oncology, Metropolitan General Hospital of Athens, 15562 Cholargos, Greece

**Keywords:** liquid biopsy, CTCs, cfDNA, mutations, NSCLC, MRD, relapse

## Abstract

**Simple Summary:**

Limited studies have focused specifically on early-stage NSCLC patients, although there is a particular clinical need for risk stratification in these patients because there are very few therapeutic options in the adjuvant setting. In this prospective study, we compared ddPCR data of hotspot mutations for four therapeutically relevant genes (*BRAF*, *EGFR*, *KRAS*, *PIK3CA*) from three different patient-matched sample types (FFPE, CTC-derived DNA and plasma-ctDNA) from early-stage NSCLC patients before surgery. We demonstrated that simultaneous analyses of plasma-cfDNA and CTC-derived DNA provided complementary molecular information from the same blood sample and greater diversity in genomic information for cancer treatment and prognosis.

**Abstract:**

Purpose: We assessed whether preoperativemutational analyses of circulating tumor cells (CTCs) and plasma-cfDNA could be used as minimally invasive biomarkers and as complimentary tools for early prediction of relapse in early-stage non-small -cell lung cancer (NSCLC). Experimental Design: Using ddPCR assays, hotspot mutations of *BRAF, KRAS, EGFR* and *PIK3CA* were identified in plasma-cfDNA samples and size-based enriched CTCs isolated from the same blood samples of 49 early-stage NSCLC patients before surgery and in a control group of healthy blood donors (*n*= 22). Direct concordance of the mutational spectrum was further evaluated in 27 patient-matched plasma-cfDNA and CTC-derived DNA in comparison to tissue-derived DNA. Results: The prevalence of detectable mutations of the four tested genes was higher in CTC-derived DNA than in the corresponding plasma-cfDNA (38.8% and 24.5%, respectively).The most commonly mutated gene was PIK3CA, in both CTCs and plasma-cfDNA at baseline and at the time of relapse. Direct comparison of the mutation status of selected drug-responsive genes in CTC-derived DNA, corresponding plasma-cfDNA and paired primary FFPE tissues clearly showed the impact of heterogeneity both within a sample type, as well as between different sample components. The incidence of relapse was higher when at least one mutation was detected in CTC-derived DNA or plasma-cfDNA compared with patients in whom no mutation was detected (*p* =0.023). Univariate analysis showed a significantly higher risk of progression (HR: 2.716; 95% CI, 1.030–7.165; *p* =0.043) in patients with detectable mutations in plasma-cfDNA compared with patients with undetectable mutations, whereas the hazard ratio was higher when at least one mutation was detected in CTC-derived DNA or plasma-cfDNA (HR: 3.375; 95% CI, 1.098–10.375; *p* =0.034). Conclusions: Simultaneous mutational analyses of plasma-cfDNA and CTC-derived DNA provided complementary molecular information from the same blood sample and greater diversity in genomic information for cancer treatment and prognosis. The detection of specific mutations in ctDNA and CTCs in patients with early-stage NSCLC before surgery was independently associated with disease recurrence, which represents an important stratification factor for future trials.

## 1. Introduction

Lung cancer (LC) is the leading cause of cancer death worldwide [1], with a 5-year survival rate of less than 20% because of its late stage at diagnosis [2]. In addition, approximately 45% of operable patients with early-stage NSCLC develop local or distant recurrence within 8–18 months due to the early subclinical spread of tumor cells.

Despite the spatial and temporal heterogeneity of tumors [3], which has been blamed for increased failure rates in targeted treatments, tissue biopsy remains the most commonly used method for tumor categorization and therapeutic decision making. Due to technological advances in identifying actionable driver mutations, several targeted therapies are available, and molecular testing for targeted genetic alterations is now considered a routine procedure [4,5,6]. In addition, liquid biopsy (LB) has attracted considerable interest in oncology as a new way to monitor tumor genetics and tumor dynamics, proving to be an important tool in the detection of minimal residual disease (MRD), which is clearly related to tumor recurrence [7]. The use of LB to detect tumor components in peripheral blood, such as circulating tumor cells (CTCs) and cell tumor DNA (ctDNA) in plasma, has been shown to be a minimally invasive tool for monitoring real-time tumor genomic changes [8] and response to treatment, as well as for investigating drug resistance and assessing the risk of tumor recurrence in operable early-stage cancer [9,10,11,12,13]. Τhe mutational landscape of CTCs in NSCLC patients may provide novel tumor biomarkers for early diagnosis of lung cancer [14] and reflect the emergence of clonal evolution under selective treatment pressure [15]. Very recently, it has been shown that recurrent tumors have distinct somatic alteration events compared with recurrent-free tumors and could demonstrate a variety of alterations correlated with time to recurrence [16].

An important question for liquid biopsy technologies is whether the composition of CTCs and plasma-cfDNA is representative of the patient’s tumor and whether they provide comparable or complementary information [17]. The main objective of the present study was to investigate whether mutational analysis of bothCTCs and ctDNA could provide a tool to detect early subclinical tumor cell dissemination, particularly before any therapeutic intervention, including surgical resection of the primary tumor, and to evaluate the risk of disease recurrence.To achieve this, we performed a pilot study, based on the direct comparison of detectable somatic hotspot mutations of four therapeutically relevant genes (*BRAF*, *EGFR*, *KRAS*, *PIK3CA*) in DNA obtained from size-based CTC fractions and matched plasma-ctDNA in patients with early-stage operable NSCLC. 

## 2. Materials and Methods

### 2.1. Study Design

A multicenter, single-arm, non-randomized, prospective observational translational research program was conducted in Greece during the period 2017–2021, evaluating the relevance of liquid biopsy (in both ctDNA and CTCs) as a dynamic biomarker for early subclinical tumor cell dissemination (minimal residual disease; MRD) and tumor evolution. Consecutive patients were enrolled in the Department of Thoracic Surgery of “Sotiria” General Hospital and were treated and followed up over time in the Departments of Medical Oncology of University General Hospital of Larissa and Metropolitan General Hospital in Athens, respectively. Eligible patients had to be ≥18 years, have a preoperative histological diagnosis of NSCLC (adenocarcinoma or squamous cell carcinoma) and a clinical stage of IA-IIIA. Clinical stage was assessed preoperatively using CT scans of the thorax, upper and lower abdomen, as well as CNS and PET/CT scans; patients with inoperable N2 or Ν3 disease, as documented by PET/CT scan and trans-bronchial lymph node biopsy guided by ultrasonography (EBUS-TBNA), were excluded from the study. Patients provided a written informed consent to participate in the study, which was approved by the Ethics and Scientificεις Committees of the participating centers “Sotiria” and “Metropolitan General” Hospitals (ref: 1288/17-1-2018 and 308/28-12-2017, respectively).

Τhe clinical samples of this study consisted of (a) formalin-fixed paraffin-embedded (FFPE) tissue from the primary tumor of 27 patients. All tissue samples were assessed microscopically in order to define the quality and cellularity (at least 70% tumor cells) of the material; (b) peripheral blood (PB), which was used for the isolation of plasma-cfDNA and CTC-derived DNA, collected before surgery (*n* = 49) and at the time of recurrence (*n* = 9); and (c) peripheral blood from health donors (HD; *n* = 22) in order to be used as controls for plasma-cfDNA and CTC-derived DNA. Peripheral blood from the patients was obtained before surgery and before the initiation of any systemic treatment. The outline of the study design is represented in Figure 1.

### 2.2. Patients Characteristics

A total of 49 patients with operable stage I-IIIA NSCLC were enrolled: 28 with squamous cell carcinoma (SCC), 18 with adenocarcinoma (ADC) and 3 with undifferentiated (NOS) carcinoma. The clinicopathologic characteristics of the enrolled patients are presented in Table 1. Most patients (84%) were smokers or ex-smokers. All patients were treatment-naïve, and 29 (59%) patients received adjuvant chemotherapy, whereas six (12%) of them received adjuvant radiotherapy according to the national and international guidelines (Appendix A). Moreover, there was no association between the presence of mutation in plasma-cfDNA or CTCs and the tumor histology, tumor size, lymph node involvement and the pathologic stage of the disease (Table 1)

### 2.3. Isolation of Genomic DNA from NSCLC Primary Tissues (FFPE)

Genomic DNA (gDNA) was isolated from primary tumor tissue sections (FFPE, 5Χ6μ) using the RNeasy FFPE Kit (Qiagen, Hilden, Germany), according to the manufacturer’s protocol.

### 2.4. Isolation of Plasma-cfDNA

Peripheral blood (20 mL) was collected in tubes containing ethylenediaminetetraacetic acid (EDTA) as anticoagulant, after discarding the first 5 mL of blood draw to avoid contamination of skin epithelial cells. Blood samples were centrifuged at 530× *g* for 10 min at room temperature (RT) and plasma was separated from buffy coat and erythrocytes. Plasma samples were then subjected to a second centrifugation at 16,000× *g* for 10 min at RT and transferred to a new tube. Aliquots of identical plasma samples from every single blood sampling were kept at −80 °C prior to cfDNA extraction. Buffy coat and erythrocytes were further processed for CTC enrichment as thoroughly described below. ctDNA was extracted from 4.00 mL of plasma using the QIAamp^®^ Circulating Nucleic Acid kit 50 (Qiagen, Hilden, Germany), as previously described [18]. 

### 2.5. CTCs Enrichment Using a Size-Based Microfluidic Device

The microfluidic device Parsortix (ANGLE plc, Guildford, UK) [19] was used for the isolation of CTCs from 20 mL whole blood. The separation of blood components took place in a microscope-slide-sized disposable cassette, which contained a series of steps, leaving a 6.5 μm measuring gap between the top cover and the final step. Following enrichment, CTCs were harvested in a total volume of 200 μL of PBS deposited into 1.5 mL Eppendorf tubes by applying reverse flow to the cassette using a specific software protocol [20,21,22]. The isolation of gDNA from enriched CTCs was performed using TRIZOL-LS (Thermo Fisher Scientific, Waltham, MA, USA) as previously described [23,24].

### 2.6. Whole-Genome Amplification

A total of 3–5 µL of gDNA (more than 1 ng) extracted from the enriched CTC fractions was amplified using the Ampli1™ Whole Genome Amplification (WGA) kit (Menarini Silicon Biosystems, Castel Maggiore BO, Italy). The WGA procedure was optimized to use gDNA derived from CTC fraction, and the protocol steps were performed according to the manufacturer’s instructions, except for the lysis step, which was already performed for gDNA extraction using TRIZOL LS.

### 2.7. Quality Control

Positive and negative controls were used in all steps to ensure the quality and reproducibility of the results. DNA integrity of all DNA samples was checked by amplification of a region in exon 20 of the *PIK3CA* gene, as described previously [25]. The quality of amplified DNA from CTCs was also checked by amplification of the same region of the *PIK3CA* gene using real-time qPCR. Two negative controls were included in each experimental procedure, one as a negative control of the qPCR reaction (blank) and the second as a wild-type control that contained the wild-type as a DNA template. Synthetic gene fragments or specific cell lines were used as positive controls.

### 2.8. Droplet Digital PCR Assays

Sensitive and specific droplet digital PCR assays for hotspot mutations of *BRAF, KRAS, EGFR* and *PIK3CA* were used. Specifically, drop-off ddPCR assays, as previously described [26], were used to screen for eight mutations in exon 19 of the *EGFR* gene (c.2237_2255, c.2235_2249, c.2240_2257, c.2235_2252, c.2237_2254, c.2239_2248) and four mutations in exon 2 of the *KRAS* gene (c 38G > A,C,T (pG13 > D,A,V), c 37G > A,C,T (pG13 > S,R,C), c 34G > A,C,T (pG12 > S,R,C), c 35G > A,C,T (pG12 > D,A,V)), whereas commercially available kits were used to detect three mutations in codon 600 of the *BRAF* gene (V600E (c.1799T > A, c.1799_1800del, c.1799_1800delinsAA), V600K (c.1798_1799delinsAA), V600R (c.1797_1799delinsGAG, c.1798_1799delinsAG, c.1798_1799delinsCG) (ddPCR™ BRAF V600 Screening Kit), and two hotspot mutations of *PIK3CA* (E545K and H1047R; Bio-Rad) were used. The input volume of the DNA sample was determined according to its concentration to use 10 ng DNA per reaction. DNA concentration was determined using the Qubit™ fluorometer (ThermoFisher Scientific, Waltham, MA, USA). Dilutions of DNA samples were performed when necessary. Diagnostic specificity of each assay and cut-off values were evaluated by analyzing the size-based CTC fraction and the corresponding plasma-cfDNA isolated from 22 HD and analyzed using exactly the same procedures (control group). All experiments were performed in duplicate.

### 2.9. Statistical Analysis

Statistical analysis was performed with SPSS Statistics 26.0 (IBM Corp., Armonk, NY, USA). There was no specific hypothesis to test the above considerations; therefore, there was no specific statistical model to predict the sample size required for this follow-up. The chi-squared test for independence and the Mann–Whitney test were used to compare the different groups. Recurrence-free survival (RFS) was defined as the time from surgery to the first documented recurrence of disease. The distributions of survival were estimated using the Kaplan–Meier method and compared between groups using the log-rank test. All statistical tests were two-sided, and *p* values < 0.05 were considered statistically significant. The COX proportional-hazards model was used to analyze possible factors affecting the recurrence of patients. Model variables included smoking, tumor size, pathological type, stage, chemotherapy or radiotherapy and the presence of mutation in plasma-cfDNA or CTCs. Differences and associations were considered significant when *p*< 0.05.

## 3. Results

### 3.1. Mutational Analysis of Plasma-cfDNA and CTCs before Surgery

Plasma-cfDNA was analyzed for the selected mutations by ddPCR in all 49 peripheral blood samples that were collected from early-stage NSCLC patients before surgery. At baseline, at least one gene was found to be mutated in 12/49 (24.5%) plasma-cfDNA samples. The most frequently mutated gene was *PIK3CA* (14.2%), as 6/49 (12%) harbored E545K mutation and 2/49 (4.1%) H1047R, whereas in one sample, both mutations were detected. BRAF and EGFR mutations were detected in 4.1% of the samples, while *KRAS* was detected in 2% of these 49 samples at baseline (Figure 2a). In one patient, (Pt#30) both *BRAF* and *EGFR* mutations were detected (Figure 2a); as expected, no mutations were detected in plasma-cfDNA from the cohort of HD subjects.

In addition, CTCs were isolated for all patients using the same blood samples. A total of 49 matched CTC-derived gDNA samples were analyzed for the detection of the same mutations by ddPCR after WGA. Interestingly, mutation frequency increased in the CTC-derived DNA compared with the plasma-cfDNA. In the baseline samples, at least one gene was mutated in 19/49 samples (38.8%) of the DNA isolated from CTCs, and the most frequently mutated gene was also *PIK3CA* (13/49, 26.5%) (Figure 2b). Deletions of exon 19 in the *EGFR* gene were detected in 4/49 (8.1%), which was twice as high as in plasma-cfDNA (4.1%), whereas a three-fold higher detection rate of *KRAS* mutations (3/49, 6.1%) was observed in exon 2 (Figure 2b). Finally, mutations in exon 15 of the *BRAF* gene were detected in 2/49 (6%), whereas no mutations were detected in the cohort from HD.

However, combined analyses of plasma-cfDNA and CTC-derived DNA increased this number to 53% (26/49) (Figure 2c), demonstrating that combined analyses of both these liquid biopsy components provide complementary information from the same blood sample. However, there was no correlation between mutation detection in gDNAs derived from CTCs and from plasma in the baseline samples, as 15 (30.6%) patients had detectable mutations in their CTC-derived DNA but not in plasma-cfDNA, and 6 samples (12.2%) had detectable mutations in plasma-cfDNA but not in CTC-derived DNA in the same blood collection (*p* = 0.532). Only in two samples were identical mutations in plasma-cfDNA and CTC-derived DNA detected, suggesting that the combined approach of both methods improves the accuracy of diagnosis and monitoring of tumor progression.

### 3.2. Direct Comparison of Selected Mutations Detected in Primary Tissue, Plasma-cfDNA and CTC-Derived gDNA before Surgery

Direct comparison of the mutation status of selected drug-responsive genes in CTC-derived DNA, corresponding plasma-cfDNA and paired primary FFPE tissues was performed in 27 early-stage NSCLC patients before surgery. The frequency of tested mutations observed in plasma-cfDNA and CTC-derived DNA was 15% and 30%, respectively; the gene most frequently mutated was *PIK3CA*. Similar to FFPE analysis, the most frequently altered gene was again *PIK3CA* as mutations in this gene were detected at a frequency of 18.5%. In two samples (7.4%), mutations were detected from either plasma-cfDNA or CTC-derived DNA, or both, that were concordant with FFPE. Separately, while 59% of the samples had some information in one component or the other, 7% of any alteration could be confirmed in the tissue sample. The relationship between different materials is described in Figure 3, which clearly shows the impact of heterogeneity both within a sample type and between different sample components.

Among 27 patients tested, 20 (74%) were still without progression of disease. A total of 15 out of these 20 patients (75%) were negative for the selected mutations in plasma-cfDNA and corresponding CTC-derived DNA. Six of the tested patients experience disease progression, and it is important to mention that in three of them (50%), mutations were detected only in CTCs or cfDNA (at baseline) but were not detected in tissue samples (Pt#8, Pt#19, Pt#29); Pt#18 and Pt#29 have the same characteristics in terms of type, stage and size of tumor, namely adenocarcinoma/ IIIA/ T2a, while Pt#8 referred squamous cell carcinoma/IIIA/T2b.Finally, in Pt#34, which relapsed, *EGFR* mutations were detected in both CTC-derived DNA and primary tissue (Figure 3). 

### 3.3. Mutation Analysis of cfDNA and CTC-Derived DNA at Disease Recurrence

A total of 17 (34.7%) patients experienced disease recurrence during the median follow-up period of 12 months (range 5–35); there were 9 recurrences (47.4%) in the 19 patients with preoperatively detectable gene mutations in CTC-derived DNA compared with 8 (66.7%) recurrences in the 12 patients with detectable ctDNA. Table 2 shows the comparison of mutations detected in plasma-cfDNA before surgery and at recurrence for nine patients whose peripheral blood samples were available at the time of recurrence.With the exception of Pt#6, all other patients in Table 2 received adjuvant chemotherapy after surgery.

In four patients (P#6,17,19,20), no mutations of the four analyzed genes could be detected either preoperatively or at the time of recurrence. In two patients (P#1 and P#8), E545K mutation of *PI3KCA* was detected in the preoperative sample but not in the recurrence sample, whereas in another patient (P#11), the same mutation was detected in the recurrence sample but not in the preoperative sample; in one patient (P#1), both *PI3KCA* mutations (E545K and H1045R) were detected in the preoperative sample, whereas one *EGFR* mutation (del19) was detected in the recurrence sample (Table 2).

A similar comparison of CTC-derived DNA mutations before surgery and at recurrence revealed that there were four patients in whom no mutations were detected in both samples (P#9,11,20,21), whereas two other patients (P#1 and P#8) were found to have the same mutation in both samples (Table 2). Of the remaining three patients, two had detectable *EGFR* mutations in the preoperative sample but not in the recurrent sample, while in one patient (P#19), *EGFR, KRAS* and H1047R mutations were detected in the preoperative sample, whereas a *PIK3CA* (E545K) mutation was detected in the recurrent sample.

Combined analyses of the mutations detected in plasma-cfDNA and CTC-derived DNA revealed that P#1 and P#8 had the same mutations (H1047R and E545K) preoperatively and at recurrence (Table 2).

### 3.4. Clinical Outcome According to the Mutational Landscape of Plasma-cfDNA and CTC-Derived gDNA

After a median follow-up of 33 months (range 5–54), 17/49 (34.7%), patients developed PD, and eight of them (47.0%) died. Patients with detectable mutations in plasma-cfDNA had significantly different RFS (*p* =0.034; Figure 4a) compared with those with undetectable mutations; conversely, there was no difference in terms of RFS between patients with detectable and undetectable mutations in CTC-derived DNA (27.4 and 31.3 months, *p* =0.370 Figure 4b). It is worth noting that the incidence of relapse was higher when at least one mutation was detected in CTC-derived DNA or plasma-cfDNA compared with patients in whom no mutation was detected (*p* =0.023, Figure 4c). Univariate analysis showed a significantly higher risk of progression (HR: 2.716; 95% CI, 1.030–7.165; *p* =0.043) in patients with detectable mutations in plasma-cfDNA compared with patients with undetectable mutations, whereas the hazard ratio was higher when at least one mutation was detected in CTC-derived DNA or plasma-cfDNA (HR: 3.375; 95% CI, 1.098–10.375; *p* =0.034). Multivariate analysis revealed that the detection of at least one mutation in plasma-cfDNA emerged as an independent prognostic factor for decreased RFS (HR = 3.990; 95% CI, 1.31–12.133, *p* = 0.015, Appendix A).

## 4. Discussion

Several studies have already demonstrated the clinical relevance of identifying genetic mutations in primary tumors and plasma-cfDNA in patients with advanced-stage NSCLC; moreover, detection of CTCs has been shown to be associated with poor patient clinical outcome in both advanced and early-stage NSCLC [27,28]. However, limited studies have focused specifically on early-stage NSCLC patients, although there is a particular clinical need for risk stratification in these patients because there are very few therapeutic options in the adjuvant setting. In this prospective study, we compared ddPCR data of hotspot mutations for four therapeutically relevant genes (*BRAF, EGFR, KRAS, PIK3CA*) from three different patient-matched sample types (FFPE, CTC-derived DNA and plasma-ctDNA) from early- stage NSCLC patients before surgery. Patients with available quality tissue specimens from diagnostic biopsies were isolated and compared. We demonstrated that simultaneous analyses of plasma-cfDNA and CTC-derived DNA provided complementary molecular information from the same blood sample and greater diversity in genomic information for cancer treatment and prognosis.

The prevalence of detectable mutations of the four tested genes was higher in CTC-derived DNA than in the corresponding plasma-cfDNA (38.8% and 24.5%, respectively). This result is in contrast to other studies, which show a higher mutation frequency in plasma samples [12,29,30]. It should be mentioned that the lower detection rate of mutations in plasma-cfDNA might be due to the early stage of the disease [31,32], while the isolation of CTCs was performed using an EpCAM-independent method that enriched the detection of CTCs, regardless of their epithelial or non-epithelial phenotype.

Interestingly, the most frequently mutated gene in both plasma-cfDNA and CTC-derived DNA, as well as in FFPE, was the *PIK3CA* gene, which was suggested to be one of the subclonal driving alterations in early-stage NSCLC tumors by Jamal-Hanjani et al. [33] using multi-region whole-exome sequencing. Mutations in the PI3K-AKT-mTOR pathway have previously been found to have a higher proportion of subclonal mutations than genes associated with the RAS-MAPK-ERK pathway [34]. However, previous studies have reported different mutation profiles of plasma and CTC-derived DNA, which could be attributed to the biological differences in their origins [29,30] as plasma-cfDNA is a degradative product of catabolism, present in the extracellular compartment; therefore, it is susceptible to multiple processes of damage in plasma, whereas genomic DNA fragments are released from all tumor sites. In contrast, circulating tumor cells are clearly related to the disease process, predicting more aggressive disease and increased metastasis. CTCs reflect the mobile metastatic subset of tumor cells.

There is evidence that some targetable driver alterations are clonal and occur early in tumorigenesis, whereas others are subclonal and the tumor acquires them later in evolution [33,35]. An interesting finding in the current study was that in some patients, the same gene mutation in plasma-cfDNA and CTC-derived DNA was not detected either before surgery or/and at relapse. This observation may be due to the clonal heterogeneity of NSCLC, as previously reported. For example, Chen et al. [36] reported that the overall concordance rate between tumor tissue DNA and matched plasma DNA mutations in early-stage NSCLC by targeted sequencing was 50.4%. Moreover, Chemi et al. [31] demonstrated that there was a higher number of shared mutations between CTCs and tissue obtained from a metastatic site compared with CTCs and tissue from the primary tumor when WES was used for the analysis of six single CTCs from a patient with early-stage NSCLC; moreover, the authors highlighted a unique aspect of the molecular analysis of CTCs using targeted deep sequencing because gene mutations could be detected in CTCs but not in plasma-cfDNA samples and primary tissue [37]. These observations could be attributed to several reasons, as follows: (a) single tissue samples presenting intra-tumor heterogeneity may be insufficient for identifying the complete genomic landscape of the tumor [38]; (b) gene mutations possibly restricted to single lesions may remain undetected in LB due to sampling error; (c) some subclonal mutations may have not been detected at the current sequencing depth; or (d) tumor volume may be a limiting factor because it has been reported that there is a clear association between the size of the primary tumor and the detection of plasma-ctDNA [39]. However, in the current study, we could not demonstrate a correlation between the detection of either ctDNA or CTCs and tumor size, taken at a limit of 5 cm, or the pathological stage of the disease.

To the best of our knowledge, the clinical relevance of mutations detected in both CTC-derived DNA and plasma-cfDNA in patients with early-stage NSCLC has not been clarified. According to our findings, patients without any mutation in both blood components had a significantly better median RFS compared to patients who had at least one detectable mutation. The same finding was revealed for the detection of mutations in plasma-cfDNA, which is in agreement with the findings of Chen et al. [16], who performed deep 457 gene-targeted NGS on 55 plasma cfDNAs of stage I NSCLC patients and reported a significant association between cfDNA detection and decreased DFS (*p* = 0.02) [16]. Additionally, based on the findings of Walcken et al. [40], ctDNA detection soon after tumor resection identifies patients at risk for relapse. Finally, in a multicenter, single-arm, phase II trial that enrolled 46 stage IIIA NSCLC patients demonstrated that low ctDNA levels (<1% MAF) at baseline had significantly improved PFS and OS compared to patients with high ctDNA levels using NGS panel testing (52 genes) [41].

Further controlled studies with a higher number of patients are required to confirm these observations and validate the use of the molecular landscape of plasma-cfDNA and CTC-derived DNA in the preoperative staging of operable NSCLC patients. Future challenges include trying to incorporate the routine evaluation of other rare targetable mutations in the assessment of liquid biopsy components of early-stage tumors, even considering that some of them could be quite rare in this setting and the inherent difficulties in designing specific clinical trials.

## Figures and Tables

**Figure 1 cancers-15-01877-f001:**
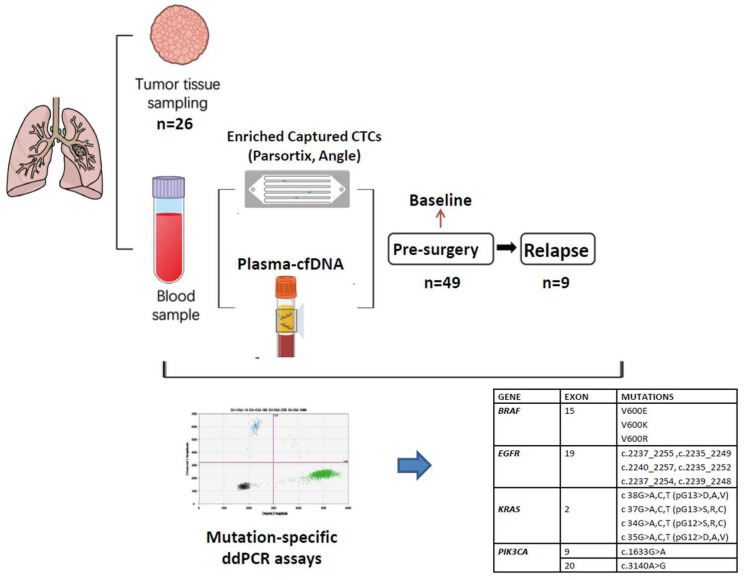
Outline of the experimental procedure.

**Figure 2 cancers-15-01877-f002:**
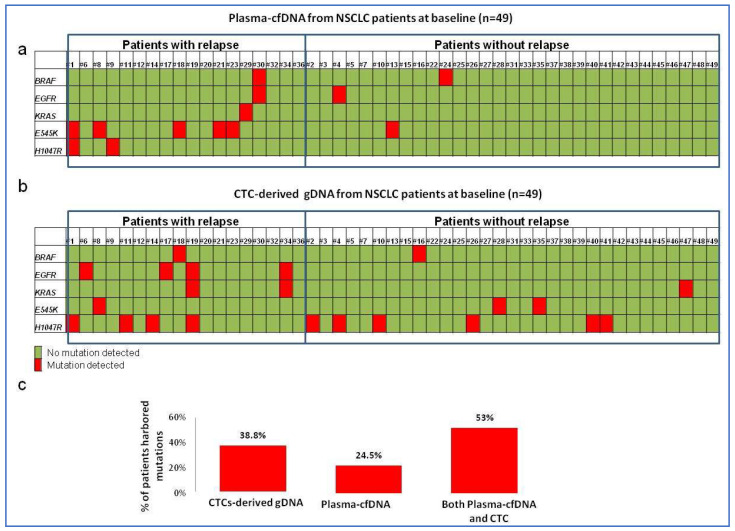
Mutational status of selected genes in (**a**) CTC-derived DNA of early-stage NSCLC patients (*n* = 49); (**b**) plasma-ctDNA of early-stage NSCLC patients (*n* = 49); and (**c**) the percentage of patients exhibiting mutations using plasma-ctDNA and CTC-derived DNA, or by analysis of plasma-ctDNA and CTCs together.The red color represents the mutation, while the green color represents WT.

**Figure 3 cancers-15-01877-f003:**
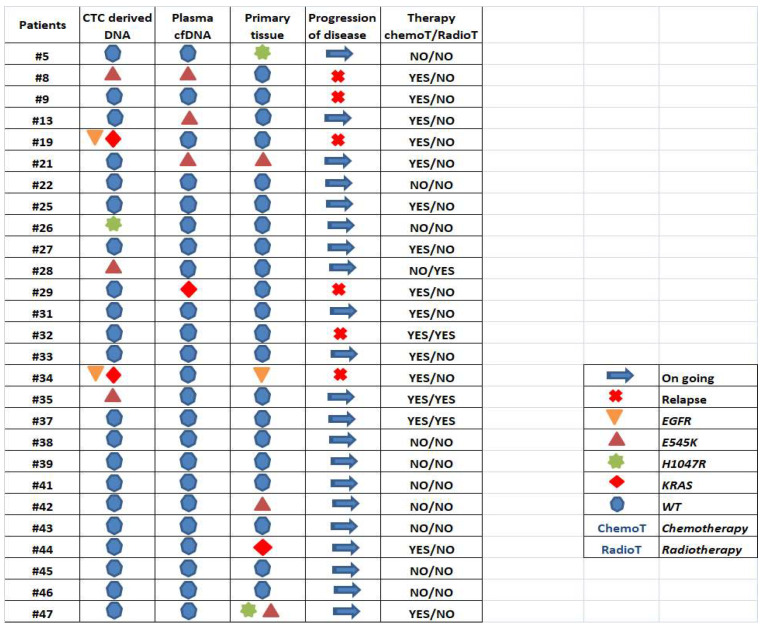
Direct comparison of tested mutations detected in primary tissue, plasma-cfDNA and CTC-derived gDNA.

**Figure 4 cancers-15-01877-f004:**
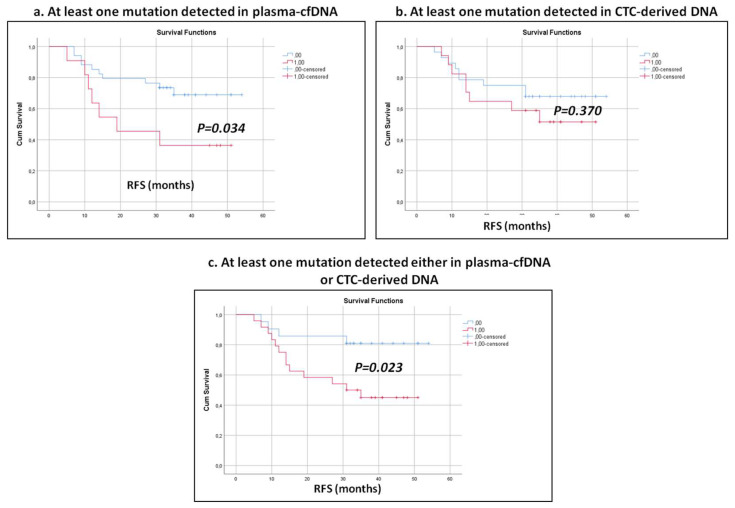
Kaplan–Meier estimates of recurrence-free survival (RFS) in months for early-stage NSCLC patients with respect to at least one mutation detected in (**a**) plasma-cfDNA, (**b**) CTC-derived DNA and (**c**) both liquid biopsy components.

**Table 1 cancers-15-01877-t001:** Clinicopathological characteristics of patients (*n* = 49).

		Presence of Mutations	
	No of Patients	cfDNA	CTCs
Average age (range)	67.7 (39–86)		
Gender			
Male	34		
Female	15		
Smoking History			
Current	21	5	10
Former	20	5	6
Never	8	0	1
		*p* = 0.691	*p* = 0.402
Histology			
Adenocarcinoma	18	3	6
Acinar Lepidic Solid Unknown	8		
	2		
	6		
	2		
Squamous Cell Carcinoma	28	9	11
Other	3	0	1
		*p* = 0.264	*p* = 0.232
Pathological Stage of lung cancer			
I	26	4	9
II-IIIA	23	8	8
		*p* = 0.193	*p* = 0.549
Tumor Size			
T1a, T2a	34	9	10
T1b, T2b	15	3	8
		*p* = 0.511	*p* = 0.103
Lymph nodes			
N0	32	5	11
≥N1	17	5	5
		*p* = 0.271	*p* = 0.516

**Table 2 cancers-15-01877-t002:** Comparison of mutations detected in (a) plasma- cfDNA isolated from plasma before surgery and at disease recurrence (*n* = 9) and in (b) CTC-derived gDNA before surgery and at disease recurrence (*n* = 9).

	Plasma-ctDNA		CTC-Derived DNA	
Patient’s Code	Before Surgery	Recurrence	Before Surgery	Recurrence
#1	**H1047R/E545K**	**EGFR**	**H1047R**	**H1047R**
#6	Undetectable	Undetectable	**EGFR**	Undetectable
#8	**E545K**	Undetectable	**E545K**	**E545K**
#9	**H1047R**	Undetectable	Undetectable	Undetectable
#11	**H1047R**	**E545K**	Undetectable	Undetectable
#17	Undetectable	Undetectable	**EGFR**	Undetectable
#19	Undetectable	Undetectable	**EGFR/KRAS/H1047R**	**E545K**
#20	Undetectable	Undetectable	Undetectable	Undetectable
#21	**E545K**	**E545K**	Undetectable	Undetectable

## Data Availability

All data generated or analyzed during this study are included in this published article (and its Appendix A).

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
