# Peer review of "Preoperative Mutational Analysis of Circulating Tumor Cells (CTCs) and Plasma-cfDNA Provides Complementary Information for Early Prediction of Relapse: A Pilot Study in Early-Stage Non-Small Cell Lung Cancer"

_cancers, 2023, doi:10.3390/cancers15061877_

Round 1
Reviewer 1 Report
Markou and colleagues present data on an evolving important prognostic clinical utility of liquid biopsy in resectable stage lung cancers. Their study data is very well written and presented.
The two biggest issues are:
1.) There is no mention in the background or discussion of plasma NGS studies used in this setting. Provencio et al J Clin Oncol 2022 demonstrated plasma NGS testing at baseline before neoadjuvant chemo-immune treatment was prognostic of outcome (and repeat plasma NGS after the neoadjuvant chemo-immune treatment was predictive of improved survival outcomes). Plasma NGS panel testing would provide broader and more complete molecular results than limited ddPCR. The prognostic proof of principle from the small ddPCR panel is still very worthy of publication but needs a fuller perspective discussion.
2.) I completely agree with the clinical utility of liquid biopsy in the pre-operative lung cancer setting, however, I have difficulty accepting the conclusion that both plasma-cfDNA and CTCs are independently associated with disease recurrence. Figure 4a confirms plasma-cfDNA is significantly associated with RFS but figure 4b in CTC-derived DNA is not significantly associated with a different RFS than no mutations. With the small numbers of the study, figure 4c seems inherently expected driven by the plasma-cfDNA prognostic value.
A minor typo is the reference superscript 41 is just 4 in the manuscript.
Author Response
Markou and colleagues present data on an evolving important prognostic clinical utility of liquid biopsy in resectable stage lung cancers. Their study data is very well written and presented.
The two biggest issues are:
1.) There is no mention in the background or discussion of plasma NGS studies used in this setting. Provencio et al J Clin Oncol 2022 demonstrated plasma NGS testing at baseline before neoadjuvant chemo-immune treatment was prognostic of outcome (and repeat plasma NGS after the neoadjuvant chemo-immune treatment was predictive of improved survival outcomes). Plasma NGS panel testing would provide broader and more complete molecular results than limited ddPCR. The prognostic proof of principle from the small ddPCR panel is still very worthy of publication but needs a fuller perspective discussion.
We would like to thank the reviewer for this so important comment. In the revised manuscript we have mentioned this point and we have added as reference 42 the suggested study.
2.) I completely agree with the clinical utility of liquid biopsy in the pre-operative lung cancer setting, however, I have difficulty accepting the conclusion that both plasma-cfDNA and CTCs are independently associated with disease recurrence. Figure 4a confirms plasma-cfDNA is significantly associated with RFS but figure 4b in CTC-derived DNA is not significantly associated with a different RFS than no mutations. With the small numbers of the study, figure 4c seems inherently expected driven by the plasma-cfDNA prognostic value.
This is a very interesting issue arising from this comment. An ability to evaluate both blood derived templates is an innovative approach that can have significant impact with regards to prognosis and on our capacity to better understand the various biological process driving metastases and potentially different therapeutic approaches. In our study it was clearly that plasma-cfDNA is significantly associated with RFS whereas the prevalence of detectable mutations was higher in CTC-derived DNA. This is why we decided to evaluate the prognostic significance firstly in each component separately and then in both components. Moreover, it is interesting to evaluate the prognostic significance of both components in higher number of patients and in longer follow-up period.
minor typo is the reference superscript 41 is just 4 in the manuscript.
In the revised manuscript we have corrected this point
Reviewer 2 Report
Dear Аuthors,
In general, the study is of great scientific significance. In my opinion, the manuscript lacks an authors’ conclusion on their own study results.
However, I have several comments:
Line 47-48: what relapses were meant? After what kind of treatment: chemo, chemoradiotherapy, combination therapy?
Line 79: Please describe the design of the study in a schematic way for better understanding.
Line 108-109: Please provide treatment options in a table if possible. If 29 (59%) patients received adjuvant chemotherapy and 6 (12%) of them received adjuvant radiation therapy, then it cannot be said that "All patients were treatment -naïve". If patients received chemotherapy and radiation therapy, which patients were included in the study? Were samples taken before or after treatment?
Line 167: it is not entirely clear how the size of the CTC was used to analyze the diagnostic specificity of each analysis (for each sample?) and thresholds? Please specify if possible.
Line 173-174: For small samples, the Mann-Whitney test is applied, and the authors write about it below. To assess the predictive value, including sensitivity and specificity, the presumptive marker level (cut-off), it was useful to perform a ROC analysis. Why did the authors not use this analysis?
Line 187. The figure is more suitable for the study design section.
Line 205: in figure 2, I did not find the gray color indicated in the caption for this figure. In my opinion, in the figure, for clarity and a better understanding of the results, patients should be divided into groups with and without relapses
Line 222 - why is the number "6" written "six"?
Line 243: Indicate in a note or in materials and methods - which patients received chemoradiotherapy or did not receive it.
Line 249: I think it is necessary to indicate the clinical and morphological characteristics of these patients. It will be of interest to readers.
Line 253-254: What is the follow-up period for these patients?
Table 2. What treatment did the patients receive? Was the treatment the same for all patients?
Lines 262 to 279: similar to the previous remark - what kind of treatment did the patients receive?
Line 326-327: It is not clear what biological differences are meant
Reference 29: «Discordant genotypes between tumor biopsy and blood-based analyses may result from technological differences, as well as sampling different tumor cell populations». That is, differences in the selection of "different populations", since the tumor is heterogeneous, as well as CTCs derived from the tumor.
Reference 35 : The authors point out that the use of multiple targets from the same tube from the same patient can give a complete picture of changes in tumor growth and response to anticancer treatment. «Indeed, the possibility to use the same blood sample to isolate and analyze cfDNA, extracellular vesicles, and CTCs would ensure no genomic and transcriptomic information is missed, yielding a total liquid biopsy».
Best regards,

Author Response
Dear Аuthors,
In general, the study is of great scientific significance. In my opinion, the manuscript lacks an authors’ conclusion on their own study results. However, I have several comments:
Line 47-48: what relapses were meant? After what kind of treatment: chemo, chemoradiotherapy, combination therapy?
Based on the literature the main modality of treatment for early-stage non-small cell lung cancer is surgery and over the past few decades, the main post-operative treatment strategy is adjuvant chemotherapy. It is well known that adjuvant chemotherapy is mainly indicated for stage IIA and beyond with certain indications for systemic chemotherapy in stage IIA. The overall benefit of adjuvant chemotherapy would be around 5% only for stages II and III. However, despite the use of chemotherapy the risk of disease recurrence or death ranges from 45% to 76% in stages I and III respectively.
Line 79: Please describe the design of the study in a schematic way for better understanding.
In the revised manuscript we have moved Figure 1 which represents the outline of study design in the appropriate section.
Line 108-109: Please provide treatment options in a table if possible. If 29 (59%) patients received adjuvant chemotherapy and 6 (12%) of them received adjuvant radiation therapy, then it cannot be said that "All patients were treatment -naïve". If patients received chemotherapy and radiation therapy, which patients were included in the study? Were samples taken before or after treatment?
We apologize for this misleading. Plasma samples were collected before surgery. None of the enrolled patients had received therapy before surgery. 59% of patients received adjuvant chemotherapy and 6% of patients received adjuvant radiation therapy after surgery. In the revised manuscript we have clarified this point. In paragraph 2.1 we have added “Peripheral blood from the patients was obtained before surgery and before the initiation of any systemic treatment”
Moreover in revised manuscript we have added a Suppl.Table 2 provided treatment of each patient after surgery.
Line 167: it is not entirely clear how the size of the CTC was used to analyze the diagnostic specificity of each analysis (for each sample?) and thresholds? Please specify if possible.
In the present study we used an EpCAM independent technology for the isolation of CTCs namely Parsortix. CTCs are caught in the Parsortix cassette due to their larger size and lower compressibility than other blood components. This is why we mention the fraction of isolated CTCs as size-based CTC.
Using peripheral blood of healthy donors and following exactly the same experimental procedure as the clinical samples we could evaluate the diagnostic specificity of each assay whereas to evaluate the background signal and to define cut-off values in specific ddPCR assays.
Line 173-174: For small samples, the Mann-Whitney test is applied, and the authors write about it below. To assess the predictive value, including sensitivity and specificity, the presumptive marker level (cut-off), it was useful to perform a ROC analysis. Why did the authors not use this analysis?
Mann–Whitney U-test was used to analyze the statistical significance and correlation of clinical pathological features, age, size tumor etc. For the evaluation of the predictive value we performed multivariate and Kaplan Meier analysis.
ROC curve, is a graphical plot that illustrates the diagnostic ability of a binary classifier system but our study was designed to evaluate the risk of disease recurrence and to evaluate the prognosis of NSCLC patients. This is why we didn’t use this analysis in this study.
Line 187. The figure is more suitable for the study design section.
We would like to thank for this comment. Based on the reviewer suggestion in the revised manuscript we have moved figure 1 in study design section.
Line 205: in figure 2, I did not find the gray color indicated in the caption for this figure. In my opinion, in the figure, for clarity and a better understanding of the results, patients should be divided into groups with and without relapses
We apologized for this misstate WT is green color; in the revised manuscript we have corrected this mistake. Moreover based on the reviewer suggestion in the revised manuscript in Figure 2 we have divided the patients into groups with and without relapse.
Line 222 - why is the number "6" written "six"?
In the revised manuscript we have corrected this point.
Line 243: Indicate in a note or in materials and methods - which patients received chemoradiotherapy or did not receive it.
In the revised manuscript on Figure 3 we have added a column namely “therapy” showing the treatment of these patients.
Line 249: I think it is necessary to indicate the clinical and morphological characteristics of these patients. It will be of interest to readers.
This is a very interesting remark. Pt#18 and Pt#29 have the same characteristics in terms of type, stage and size of tumor: adenocarcinoma/ IIIA/ T2a, while Pt#8 was squamous cell carcinoma/IIIA/T2b. In the revised manuscript we have added this information.
Line 253-254: What is the follow-up period for these patients?
The median follow-up period for these patients was 12 months (range 5-35). In the revised manuscript we have added this information.
Table 2. What treatment did the patients receive? Was the treatment the same for all patients?
Lines 262 to 279: similar to the previous remark - what kind of treatment did the patients receive?
We would like to thank the reviewer for these comments. With the exception of Pt#6, all other patients in Table 2 received adjuvant chemo-therapy after surgery. In the revised manuscript we have included this so important information.
Line 326-327: It is not clear what biological differences are meant
Plasma-cfDNA is a degradative product of catabolism, present in the extracellular compartment and therefore is susceptible to multiple processes of damage in plasma whereas these genomic DNA fragments released from all tumor sites. In contrast, circulating tumor cells are clearly related to the disease process, predict more aggressive disease and increased metastasis. CTC reflect the mobile metastatic subset of tumor cells. In the revised manuscript we have clearly explained this point.
Reference 35: The authors point out that the use of multiple targets from the same tube from the same patient can give a complete picture of changes in tumor growth and response to anticancer treatment. «Indeed, the possibility to use the same blood sample to isolate and analyze cfDNA, extracellular vesicles, and CTCs would ensure no genomic and transcriptomic information is missed, yielding a total liquid biopsy».
We absolutely agree with this comment. Moreover in the same study the author mentioned that based on their data “Mutational profiles from cfDNA and gDNA from CTCs differ significantly and together may give a more comprehensive picture. These results show that the combination of cfDNA and CTCs may be more useful than either test alone.” This is way we added this reference in this certain point.
Reviewer 3 Report
1. Line 22 - what is n=49 pts? Decipher please.
2. There is no analysis of clinical and pathological characteristics of the tumor depending on the presence/absence of mutations in plasma-cfDNA and CTCs. Do they differ in histological type, stage, lymph node involvement, etc.?
3. Patients with relapse are not described in terms of clinical and pathological parameters of the disease. The prognosis and risk of recurrence may be related to the prevalence of the tumor process, the type of treatment, and many other factors. However, a multivariate survival analysis has not been performed. The presence/absence of mutations in plasma-cfDNA and CTCs has not been shown to be comparable to other prognostic and risk factors for recurrence.
Author Response
- Line 22 - what is n=49 pts? Decipher please.
We apologize for this misleading and clearly mentioned in the revised manuscript that "49 early-stage NSCLC patients before surgery."
- There is no analysis of clinical and pathological characteristics of the tumor depending on the presence/absence of mutations in plasma-cfDNA and CTCs. Do they differ in histological type, stage, lymph node involvement, etc.?
Thank you very much for this suggestion. In the revised manuscript we have modified Table 1 and we have added the results of correlation study of the presence of mutations in ctDNA or CTC with Clinicopathological characteristics of patients. Moreover in paragraph 2.2 we have added the following state “Moreover, there was no association between the presence of mutation in plasma-cfDNA or CTCs and the tumor histology, tumor size, lymph node involvement and the pathologic stage of the disease (Table 1)”
- Patients with relapse are not described in terms of clinical and pathological parameters of the disease. The prognosis and risk of recurrence may be related to the prevalence of the tumor process, the type of treatment, and many other factors. However, a multivariate survival analysis has not been performed. The presence/absence of mutations in plasma-cfDNA and CTCs has not been shown to be comparable to other prognostic and risk factors for recurrence.
We thank the reviewer for this so important comment. In the revised manuscript we have performed multivariate analysis and we have also added a Supplementary Table 1 showing the results that revealed.
The COX proportional hazards model was used to analyze the possible factors affecting the recurrence of patients. Model variables included smoking, tumor size, pathological type, stage, chemotherapy or radiotherapy, presence of mutation in plasma-cfDNA or CTCs (Suppl. Table 2). In multivariate analysis, reported as hazard ratio and 95% CI, the result showed that the detection of at least one mutation in plasma-cfDNA emerged was independently associated with the relapse of patients.
In the revised manuscript we have also added the following sentences in paragraph 3.4 “The multivariate analysis revealed that the detection of at least one mutation in plasma-cfDNA emerged as an independent prognostic factor for decreased RFS (HR = 3.990; 95% CI, 1.31–12.133, P = 0.015, Suppl.Table 2).”
Round 2
Reviewer 2 Report
Dear authors! Overall, your answers have been satisfactory. The manuscript has been revised according to the comments of the reviewer.
Reviewer 3 Report
The authors responded to the reviewer's comments and made the appropriate corrections. In its present form, the article can be recommended for publication.